# The Diagnostic Utility of Oligoclonal Bands in Multiple Sclerosis: A Time-Course Analysis

**DOI:** 10.3390/biomedicines13020440

**Published:** 2025-02-11

**Authors:** Raffaella Candeloro, Caterina Ferri, Michele Laudisi, Eleonora Baldi, Maura Pugliatti, Massimiliano Castellazzi

**Affiliations:** 1Department of Neurosciences and Rehabilitation, University of Ferrara, 44121 Ferrara, Italy; raffaella.candeloro@unife.it (R.C.); maura.pugliatti@unife.it (M.P.); 2Department of Neuroscience, “S. Anna” University Hospital, 44124 Ferrara, Italy; caterina.ferri@ospfe.it (C.F.); michele.laudisi@ospfe.it (M.L.); e.baldi@ospfe.it (E.B.); 3Interdepartmental Research Center for the Study of Multiple Sclerosis and Inflammatory and Degenerative Diseases of the Nervous System, University of Ferrara, 44121 Ferrara, Italy

**Keywords:** multiple sclerosis, cerebrospinal fluid, oligoclonal bands, diagnosis

## Abstract

**Background**: Multiple sclerosis (MS) is a chronic inflammatory disease of the central nervous system (CNS). Oligoclonal bands (OCBs) in cerebrospinal fluid (CSF) are a hallmark of MS and reflect intrathecal IgG synthesis and inflammation. This study aims to analyze the temporal distribution of IgG OCBs in the CSF of patients with a definitive diagnosis of MS. **Methods**: This retrospective study included 500 patients with diagnosed MS. Patients were divided into four groups according to diagnostic epochs: Group 1 (Pre-2001 or Pre-McDonald), Group 2 (2001–2010 or McDonald 2001-Polman 2010), Group 3 (2010–2018 or Polman 2010), and Group 4 (Post-2018 or Thompson 2017). Statistical analyses examined temporal and sex differences in OCB positivity rates. **Results**: OCB positivity was lower in Group 4 (69.2%) compared to Group 1 (85.4%) in the overall population (*p* = 0.0022). A decrease in OCB positivity was observed in Groups 3 (62.5%) and 4 (71.8%) compared to Group 1 (92.5%) among males (*p* = 0.0117 and *p* = 0.0198, respectively) and in Group 4 (68.1%) compared to Group 1 (82.5%) among females (*p* = 0.0274). **Conclusions**: The present study provides valuable insights into temporal trends in CSF positivity among patients diagnosed with MS. There was an overall decrease in OCB positivity rates over the years, particularly in the post-2018 period.

## 1. Introduction

Multiple sclerosis (MS) is a chronic inflammatory disease of the central nervous system (CNS) [1]. Diagnosing MS is complex and multidisciplinary, involving clinical, laboratory, and magnetic resonance imaging (MRI) examinations, as no single clinical feature or diagnostic test is sufficient for diagnosis [2]. The diagnostic criteria established for MS have undergone updates over the years; in fact, changes were made in 2001 to integrate dissemination in space and time, a key concept in the diagnosis of MS [2]. It refers to the spread of damage or inflammation within the CNS over time and across different locations. Clinical examination and MRI scans play crucial roles in assessing this dissemination. Cerebrospinal fluid (CSF) analysis, demonstrating the presence of local inflammation within the CNS, plays an important role in MS diagnosis, especially in cases in which clinical information on temporal dissemination is lacking [3]. CSF is a colorless biological fluid, similar to ultra-filtrated plasma, with a lower protein and lymphocyte content [4]. Lumbar puncture, which is defined as a minimally invasive surgical practice [5], is performed to collect CSF. Oligoclonal bands (OCBs) in cerebrospinal fluid are a hallmark of multiple sclerosis, reflecting intrathecal IgG synthesis and inflammation [6]. For decades, the “gold standard” for determining intrathecal IgG synthesis has been isoelectric focusing (IEF) on agarose gel followed by IgG-specific immunofixation of paired serum and CSF samples [7]. To minimize bias, it would be advisable to have IEF gels interpreted by two independent observers who are blinded to the clinical information. Inter-observer agreement would be used to reach a consensus [8]. The assignment of one of the five possible patterns defined in the guidelines also depends on this agreement [6]: pattern 1 = absence of IgG OCBs; pattern 2 = two or more CSF-restricted IgG OCBs; pattern 3 = two or more CSF-restricted IgG OCBs with additional identical IgG OCBs in CSF and serum; pattern 4 = two or more identical IgG OCBs in CSF and serum; pattern 5 = some identical IgG OCBs in CSF and serum in a restricted pH range. Only patterns 2 and 3 indicate intrathecal IgG synthesis [6]. The persistence of OCBs in MS is driven by long-lived plasma cells, which are highly resistant to immunosuppression [8,9,10]. Typical cellular and protein abnormalities in CSF generally remain relatively stable during MS [5].

Initially, the search for OCBs represented a fundamental point for confirming the diagnosis in suspected cases, but over the years, with the advancement of technology and the latest state-of-the-art MRI scans and the refinement of McDonald’s criteria, the importance of the bands seems to have diminished. In 2010, the revision of the criteria opted for the elimination of CSF analysis [11]; it was, however, reintroduced in 2017, using the positivity of the bands to define the diagnosis of MS [3].

Analyzing data from our study, we observed notable changes in OCB positivity rates compared to historical reports, which previously indicated a 90% positivity rate. To offer a comprehensive view of long-term trends in clinical practice, we conducted an analysis that accounted for evolving diagnostic criteria and stratified the data by sex. The aim of this study is to evaluate the temporal distribution of IgG OCBs in the CSF of patients at the time of lumbar puncture.

## 2. Materials and Methods

### 2.1. Study Design

This was an observational retrospective study which included patients at the Sant’Anna University Hospital in Ferrara, Italy, who underwent lumbar puncture and were subsequently diagnosed with MS. Due to its observational retrospective design, this study relied on existing data, reflecting a real-world cohort. It included only patients with a confirmed diagnosis of MS who had undergone lumbar punctures within the past 30 years. No new patients were enrolled. The study was approved by the Medical Ethics Committee in Research “Ethics Committee of Area Vasta Emilia Centro of the Emilia-Romagna Region” (protocol number 770/2018/Oss/AOUFe). Clinical and laboratory data were collected anonymously. Both pre-analytical and analytical procedures were performed according to good clinical practice and following international guidelines [6]. All patient data refer to the time when the lumbar puncture was performed. The study did not make any changes to the diagnosis.

### 2.2. Study Population

From an initial dataset of 2635 patients within the Laboratory of Neurochemistry at the Sant’Anna University Hospital in Ferrara, 500 patients with a definitive diagnosis of MS were identified for this study (149 males and 351 females). A total of 2135 patients were excluded due to other diagnoses or incomplete data. Patients were divided into four groups based on the year of spinal examination and the diagnostic criteria in force at that time: Group 1 (Pre-2001 or Pre-McDonald) [2], Group 2 (2001–2010 or McDonald 2001-Polman 2010) [2,11,12], Group 3 (2010–2018 or Polman 2010) [11], and Group 4 (Post-2018 or Thompson 2017) [3].

Exclusion criteria were CSF white blood cells >10/µL, the presence of CSF discolorations, and repeated CSF samples [13].

### 2.3. Sample Analysis

All analyses were conducted in accordance with good clinical practice as part of routine diagnostic procedures at the time of hospitalization. CSF and serum samples from all patients were collected and processed using standard protocols. The Helena Biosciences IgG IEF kit (Helena Biosciences, Gateshead, UK) was consistently used for all analyses. To ensure reliability, all IEF runs were independently interpreted by at least two blinded operators, following the criteria published by Andersson et al. [6]. Inter-observer agreement was reached for the definitive interpretations. The presence of intrathecal IgG synthesis was determined based on the presence of a CSF-restricted OCB (patterns 2 and 3 according to Andersson’s criteria).

CSF and serum samples were collected from all patients and centrifuged at 300× *g* for 10 min at 20 °C to separate the liquid component from the cellular portion. All samples were stored and analyzed under the same conditions.

At the time of lumbar puncture, the IgG OCB was determined in paired CSF and serum samples in all patients to establish the diagnosis, using the current “gold standard” IEF followed by IgG-specific immunoblotting. For this analysis, the SAS-3 instrument and Helena Biosciences IgG IEF kit and SAS IgG IEF kit (Helena Biosciences, Gateshead, UK) were used.

### 2.4. Statistical Analysis

All data were preliminary checked for normality using the Kolmogorov–Smirnov test. Continuous variables showing a normal distribution were presented as the mean and standard deviation (SD). Categorical variables were reported as counts (percentages), and Fisher’s exact test was used to identify differences in proportions among the four groups and subsequently to conduct pairwise comparisons between Groups 2, 3, and 4 and Group 1, both in the overall patient population and in the subgroups stratified by sex. Two-tailed *p*-values of less than 0.05 were considered statistically significant. Prism 10 software for MacOS (GraphPad Software, La Jolla, CA, USA) was used for the statistical analyses.

## 3. Results

The mean age of the total population was 38.2 years (SD = 11.7). No statistically significant differences in mean age were observed between males and females. An initial analysis of patient patterns revealed the following distribution (Table 1): (i) pattern 1: 22 males and 66 females; (ii) pattern 2: 119 males and 259 females; (iii) pattern 3: 4 males and 18 females; (iv) pattern 4: 0 males and 8 females; (v) pattern 5: no patients observed. Overall, 68.1% of the patients tested positive (pattern 2 or 3).

Over the years, the positivity rate for OCBs (pattern 2 or 3) changed according to the revisions of the McDonald criteria, as shown in Table 2.

There was a significant difference in the distribution of OCB positivity over time in the total patient population (Fisher’s exact test: *p* = 0.0019). Specifically, OCB positivity was lower in Group 4 compared to Group 1 (Fisher’s exact test: *p* = 0.0022). This temporal trend in OCB positivity was also evident in both the male (Fisher’s exact test: *p* = 0.0078) and female subgroups (Fisher’s exact test: *p* = 0.0417). Notably, a decrease in OCB positivity was observed in Groups 3 and 4 compared to Group 1 among males (Fisher’s exact test: *p* = 0.0117 and *p* = 0.0198, respectively) and in Group 4 compared to Group 1 among females (Fisher’s exact test: *p* = 0.0274). No differences were found between the two sexes in the various groups.

## 4. Discussion

The present study provides valuable insights into temporal trends in CSF positivity among patients diagnosed with MS. Our results highlight the evolving role of CSF analysis in the diagnostic process, particularly in relation to revisions of the McDonald criteria.

A key finding is the overall decrease in OCB positivity rates over the years, particularly in the post-2018 period. This decrease may reflect several factors, including the increasing use of high-resolution MRI to meet the criteria for spread in time and space, reducing the timing of diagnosis. The lack of significant differences between Group 1 (pre-2001) and Group 2 (2001–2010) in the use of OCBs could reflect the persistence of diagnostic practices based on this biomarker, which continued to be used due to its high specificity. During this period, the integration of the McDonald diagnostic criteria did not lead to an immediate change in clinical methodologies.

Immune response differences between sexes are well documented, with females generally exhibiting more robust humoral immune responses, potentially influencing the detection and prevalence of intrathecal immunoglobulin synthesis [14]. However, no statistically significant difference in OCB positivity emerged between males and females.

In this case, it can be assumed that MRI played a crucial role in identifying the disease, supporting the difference found between Groups 3 and 4 and Groups 1 and 2. Advances in MRI technology, the greater number of instruments, and the increased frequency with which patients undergo this examination, together with the development of more sensitive diagnostic criteria, have allowed earlier and more accurate diagnoses. Furthermore, the accuracy of clinical diagnosis has undoubtedly improved over time, driven by advancements in our understanding of the disease, ongoing medical education, and enhanced collaboration among clinical, radiological, and laboratory specialists. However, studies have demonstrated that relying solely or heavily on MRI examinations can lead to misdiagnosis, even in individuals who do not have inflammatory neurological diseases and would likely have negative CSF test results [15]. Therefore, OCBs remain useful in specific contexts, such as in cases where MRI does not fully meet the criteria for temporal and spatial dissemination, or in differential diagnosis contexts. It is important to highlight how the exclusion of OCBs from the diagnostic criteria for MS in 2010 [11] generated significant concerns within the scientific community. Subsequent studies have indeed demonstrated the risk of misdiagnosing MS due to the absence of CSF examination [15]. The authors have shown the risk of diagnosing non-inflammatory/autoimmune diseases as MS, a situation that would not have occurred if the search for OCBs had been conducted [15]. Today, new biomarkers are entering the context of CSF analysis, such as kappa free light chains (KFLCs); these new biomarkers seem to be able to replace the presence of OCBs. In addition, KFLCs have several strengths due to their high sensitivity and the possibility of an automated process with subsequent calculation of a mathematical index, the K-index [16,17,18]. This new method requires less time and manpower. Considering these new biomarkers, which could bring significant advantages in the diagnosis of MS, future iterations of the diagnostic criteria may aim to expedite disease recognition [18]. However, recent studies have shown that the specificity of KFLCs may vary between the two sexes, which highlights how future thresholds proposed in clinical practice must take these differences into account [19].

In 2013, Andreadou and coworkers reported an OCB positivity rate similar to our findings, with a percentage of 67.5% among CIS patients and people with a definitive diagnosis of MS, without differences between the two groups. Notably, in that cohort, OCB-positive patients were significantly younger [20].

Tertiary lymphoid organs (TLOs) seem to play an important role in the development and maintenance of intrathecal IgG synthesis in multiple sclerosis (MS) [21,22]. TLOs provide the necessary environment for B-cell maturation, clonal expansion, and affinity maturation. This process leads to the generation of high-affinity antibodies [23]. Evidence suggests that TLOs exist within the central nervous system (CNS), contributing to the local immune response and potentially driving disease progression [24]. Bonnan, in 2014, emphasized that intrathecal IgG synthesis, indicated by OCB presence, is a robust and persistent feature of MS, detected in over 95% of patients, and that this intrathecal synthesis remains relatively unchanged throughout life, even with advancements in immunomodulatory therapies [6,25]. Although natalizumab can partially and paradoxically reduce IgG synthesis, OCB persistence remains high following treatment [25].

It remains unclear whether CSF antibodies hold prognostic significance in MS or if OCB-negative patients represent a distinct disease subtype. The literature presents conflicting evidence regarding the association between disease severity and OCB presence. Some studies suggest that OCB-positive patients have a worse prognosis than OCB-negative ones [20,26,27]. More recent research found that CSF-specific OCBs were present in 87.7% of MS cases, with female patients showing a higher intensity of intrathecal IgG production [28]. Age-related differences were also observed, with positivity rates of approximately 90% in adults and 40–60% in pediatric patients [8].

The distribution of positivity also varies according to geographical area. The prevalence of IgG OCBs in the CSF of MS patients is 81–87.7% in southern European countries (Portugal, Spain), central Europe (Czech Republic), and transcontinental Turkey and exceeds 90% in the Nordic regions, the UK, and Canada [20,29,30]. In line with what has been reported in those studies, our data seem to confirm previous findings that the prevalence of OCBs is lower than the frequency found in other Western countries, particularly in Northern Europe and Canada, where rates above 90% have been reported [20,29].

OCB positivity remains an important biomarker of intrathecal IgG synthesis and is a hallmark of MS-related inflammation. However, its diagnostic significance seems to be diminishing in light of technological advances. While patterns 2 and 3 continue to indicate intrathecal IgG synthesis, the decrease in OCB positivity rates could also suggest changes in patient selection criteria for lumbar punctures or improvements in the early diagnosis of MS through imaging.

A strength of this study is its statistical analysis stratified by time period and sex. However, this study is not without limitations. Despite the overall large sample size, a limitation of this study is the size of some subgroups, which were created by dividing the cohort by sex and time period. Future studies with larger cohorts may confirm or refute the trends we observed. The exclusion of 2135 patients due to insufficient or incomplete data limits the generalizability of the results. Furthermore, this study was conducted in a single center, and the results may not be fully representative. These results suggest that although the OCB test remains a valid diagnostic tool, its role in MS diagnosis may need to be re-evaluated in light of evolving diagnostic criteria and technological advances. Clinicians should consider a more integrated approach combining clinical findings, imaging, and CSF analysis, with a nuanced understanding of sex-based differences in disease presentation. Multicenter prospective studies are also needed to validate these findings.

## 5. Conclusions

This study underscores the dynamic nature of MS diagnosis and the shifting role of CSF analysis in clinical practice. While OCB testing continues to provide critical information about intrathecal inflammation, its diagnostic utility appears to be increasingly supplemented by advances in imaging and refined diagnostic criteria. Future research should focus on integrating emerging biomarkers, such as KFLCs, with traditional methods to create a more comprehensive and personalized approach to MS diagnosis.

## Figures and Tables

**Table 1 biomedicines-13-00440-t001:** Distribution of patterns based on OCB analysis in patients with a definitive diagnosis of multiple sclerosis.

	Males n = 149	Females n = 351	*p*-Value
Age: Mean (SD)	37.9 (11.5)	38.4 (11.8)	
OCB Profiles			
Pattern 1: n (%)	22 (14.1)	66 (18.8)	0.3093
Pattern 2: n (%)	119 (76.39)	259 (73.8)	0.1720
Pattern 3: n (%)	4 (2.6)	18 (5.1)	0.3395
Pattern 4: n (%)	4 (2.6)	8 (2.3)	0.7547
Pattern 5: n (%)	0	0	>0.9999

Pattern 1 = absence of IgG OCBs; pattern 2 = 2 or more CSF-restricted IgG OCBs; pattern 3 = 2 or more CSF-restricted IgG OCBs with additional identical IgG OCBs in CSF and serum; pattern 4 = 2 or more identical IgG OCBs in CSF and serum; pattern 5 = some identical IgG OCBs in CSF and serum in a restricted pH range. Only patterns 2 and 3 indicate intrathecal IgG synthesis. Abbreviations: SD, standard deviation; OCB, cerebrospinal fluid-restricted IgG oligoclonal bands.

**Table 2 biomedicines-13-00440-t002:** Description of the change in positivity over the years, according to McDonald’s criteria.

	Group 1 “Pre-2001”	Group 2 “2001–2010”	Group 3 “2011–2018”	Group 4 “Post 2018”	*p*-Value
Total: n	137	159	74	130	
OCB+: n (%)	117 (85.4)	136 (85.5)	57 (77.0)	90 (69.2) ^a^	**0.0019**
Males: n	40	54	16	39	
OCB+: n (%)	37 (92.5)	48 (88.9)	10 (62.5) ^b^	28 (71.8) ^c^	**0.0078**
Females: n	97	105	58	91	
OCB+: n (%)	80 (82.5)	88 (83.8)	47 (81.0)	62 (68.1) ^d^	**0.0417**

Fisher’s exact test was used to compare sexes. Bold *p*-values indicate statistical significance. Abbreviation: OCB+, cerebrospinal fluid-restricted IgG oligoclonal band positivity. ^a^ Group 4 vs. Group 1: *p* = 0.0022; ^b^ Group 3 vs. Group 1: *p* = 0.0117; ^c^ Group 4 vs. Group 1: *p* = 0.0198; ^d^ Group 4 vs. Group 1: *p* = 0.0274.

## Data Availability

The datasets used and analyzed during the current study are available from the corresponding author upon reasonable request.

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
