# Peer review of "The Diagnostic Utility of Oligoclonal Bands in Multiple Sclerosis: A Time-Course Analysis"

_biomedicines, 2025, doi:10.3390/biomedicines13020440_

Round 1
Reviewer 1 Report
Comments and Suggestions for Authors
see attach

The English could be improved to express the research more clearly.
Reviewer 2 Report
Comments and Suggestions for Authors
The conclusion requires improvement. In its current form, it does not reflect the findings of the study.
How did the authors calculate the sample size?
What were the inclusion criteria?
Was there any patient recruitment for this study?
Table 1 shows no statistical analysis.
Did the authors consider any other variables (occupation, employment status, racial group, comorbidity, etc.)?
What is the clinical significance of the findings? The authors mentioned brief information about this topic.
The conclusion section should be rewritten. In its current form, the information appears to be a perspective section.
The format of some references needs to be checked and corrected (e.g., 20 and 22).
Reviewer 3 Report
Comments and Suggestions for Authors
This report presents a clinical study on the use of oligoclonal bands (OCBs) for diagnosing multiple sclerosis. The data presentation and manuscript are well-prepared. The reviewer suggests that the report can be accepted after addressing the following minor concerns:
1. The reviewer recommends that the authors emphasize the uniqueness and significance of this study. Currently, the Introduction focuses primarily on the general importance of using OCBs, which could be balanced with a stronger discussion of the study's distinct contributions.
2. The manuscript would be more engaging if the authors provided insights into the observed differences in the decrease of OCB positivity between males and females. This could enhance the understanding of potential underlying mechanisms or contributing factors.
3. The authors are encouraged to discuss the current limitations of detecting OCB positivity. For example, they could suggest promising new devices or necessary advancements in existing techniques to improve detection. This addition would make the report valuable not only for clinical researchers but also for those from engineering backgrounds.
Reviewer 4 Report
Comments and Suggestions for Authors
The manuscript by Candeloro et al. needs some clarification.
Please define the patterns in Table 1. Also, define OCB positivity. What are the differences between groups?
Discussion is somewhat not focused.
Round 2
Reviewer 1 Report
Comments and Suggestions for Authors
Authors have partially addressed the reviewer’s comments, resulting in a better-structured manuscript with some sections more thoroughly discussed. However, certain concerns remain as some elements might still be unclear to readers.
Authors claim that humoral differences in immune response between men and women influence the detection of oligoclonal bands (OCBs). However, their data contradict this assertion, as no statistically significant differences are reported.
Authors suggest that the K-index "seems to replace OCBs," but this has already been extensively demonstrated in the literature and was officially stated at the ECTRIMS 2024 conference in Copenhagen. During ECTRIMS 2024, the proposed update to the McDonald 2017 diagnostic criteria was presented, introducing the McDonald 2024 criteria. These revisions effectively supersede discussions on OCBs in favour of the K-index.
Contradictory elements remain that could create confusion among readers.
Comments on the Quality of English LanguageThe English could be improved to more clearly express the research.
Reviewer 2 Report
Comments and Suggestions for Authors
the manuscript can be accepted for publication